# Waterproofing a Thermally Actuated Vibrational MEMS Viscosity Sensor

Luis Gan, Shreyas Choudhary, Kavana Reddy, Connor Levine, Lukas Jander, Amogh Uchil and Ivan Puchades * 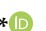

Electrical and Microelectronic Engineering Department, Rochester Institute of Technology, Rochester, NY 14568, USA; lag6597@g.rit.edu (L.G.); sc3355@rit.edu (S.C.); kavana.reddy@amd.com (K.R.); cxl4088@rit.edu (C.L.); lj5888@rit.edu (L.J.); aru5169@rit.edu (A.U.)
* Correspondence: ixpeme@rit.edu

**Abstract:** An efficient and inexpensive post-process method to waterproof an electrically actuated microtransducer has been studied. The electrical signals of microtransducers operating in electrically conductive fluids must be effectively isolated from the surrounding environment while remaining in contact for sensing purposes. A thermally actuated MEMS viscosity sensor uses electrical signals for both actuation and sensing. Three post-processing materials, (1) Parylene-C, (2) flouroacrylate-based polymer, and (3) nitrocellulose-based polymer, were coated as thin layers of waterproofing materials on different sensors. All three coating materials provided adequate protection when tested under normal operating conditions. Although the vibration response of the sensors was slightly modified, it did not affect their functionality in a significant way when measuring conductive fluids based on glycerol–water mixtures. All the treated sensors lasted over 1.2 million actuations without any decay in performance or failures. When the test bias conditions were increased by 5x to accelerate failures, the flouroacrylate-based polymer samples lasted 2x longer than the others. Visual analysis of the failures indicates that the edge of the diaphragm, which undergoes the most significant stress and strain values during actuation, was the location of the mechanical failure. This work guides post-processed waterproofing coatings for microscale actuators operating in harsh and damaging environments.

**Keywords:** MEMS; viscosity sensors; microelectronics; waterproofing

## 1. Introduction

Viscosity is one of the most critical rheological measurements of fluids. Many recent reports on miniaturized viscosity sensors offer the advantage of inexpensive, in situ, real-time, and continuous measurements [1]. These are being used in many different industries, including automotive [2,3] and industrial settings [4–8], to monitor fluid conditions. In addition, biomedical applications also use viscosity sensors to monitor things like blood coagulation rates and other biological properties of fluids [9–12]. For example, the frequency shift of a piezoelectric AlN transducer is used in [5] to measure the fermentation of the process in wine, technical lubricants, and a highly viscous bitumen liquid (64,000 cP). In [9], a continuous microfluidic viscometer with no moving parts measures blood coagulation rates as the viscosity increases from 5 to 100 cP. In [13], a miniaturized tuning-fork-based mechanical oscillator is used in oil and gas applications to measure viscosity in the range of 40 to 100 cP, as well as density.

Many of these miniaturized sensors are fabricated using a micro-electro-mechanical (MEMS) fabrication process. Although many different types of actuation and sensing are reported, this paper will focus on those based on vibrating elements. These vibrating sensors measure viscosity by using cantilever beams or plates [14–17], resonating membranes [18], or through quartz crystal resonators technology [19]. These vibration elements must be in contact with the fluid being tested to measure their rheological properties effectively.

The vibration elements in MEMS sensors are generally actuated through an electrical signal, and their behavior and interaction with the fluid are measured through another electrical signal. Both of these electrical signals are generally on-chip and integrated within the sensor. These electrical signals need to be effectively isolated from one another to obtain a reliable measurement. Isolating the electrical signals from the fluid is generally not a concern when measuring nonelectrically conductive fluids such as lubricating oils used in automotive or industrial settings. On the other hand, an effective and reliable way to isolate the on-chip electrical signals is needed when measuring electrically conductive fluids in the food industry or biomedical applications.

There are several methods to waterproof electronics by applying a hydrophilic film. These include the CVD-based Parylene-C [20–25], silicone-based coating, and other polymers [26–28]. Even though these have been proven effective at protecting packaged electronics and components placed on a PCB [28–31], few studies have shown their effectiveness when protecting actuators embedded in the sensing environment [25].

As such, there is a need to adequately protect the electronic signals of MEMS devices, including sensors and micro-actuators submerged in fluids or other harsh environments. This work briefly describes the fabrication and packaging of thermally actuated MEMS viscosity sensors. Several sensors are tested in oil before applying three different waterproofing materials. The sensors are tested for differences in sensing performance in a conductive fluid before testing their longevity in fluids with an accelerated bias. Finally, an analysis of the failures is described to conclude this work, which guides coating solutions for microscale actuators operating in harsh and damaging environments.

## 2. Materials and Methods

Thermally actuated MEMS viscosity sensors were fabricated using the process described in [18]. After fabrication, the sensors are singulated using a wafer saw and attached to an FR-4 PCB with a two-part epoxy (Loctite Epoxy Heavy Duty). The PCB has an access hole drilled on the back so that the fluid under test can access both sides of the membrane. The sensor is then wire-bonded to the copper traces in the PCB. The wire-bonding wire (75 µm diameter Al/Si wire) and the metal pads on the sensor are then covered with the same two-part epoxy for electrical isolation and protection during handling and testing. The center of the membrane is exposed to the liquid. Lastly, pin headers are soldered to the PCB board a distance away from the sensor.

The fabricated sensors are tested in standard calibrating oils N10, N26, N35, and N100 (Cannon Instruments Company, State College, PA, USA). The sensors are actuated with a 20 V, 50 µsec pulse at a repetition rate of 5 Hz. The heater resistance is 100 Ohms. The resulting signal corresponds to a free response of an underdamped harmonic oscillation and is monitored and measured using a LabVIEW interface, as described in [29]. Data are collected for 3 min in each oil. The sensors are cleaned with a degreaser solution (Formula 409), rinsed in DI water and isopropyl alcohol (IPA), and thoroughly dried between tests.

One packaged sensor without the waterproofing coat is tested with the same test bias conditions (20 V, 50 µsec pulse at a repetition rate of 5 Hz) until failure. The failure is inspected with optical and scanning electron microscopes (SEM).

Three waterproofing materials are then applied to different sensors: Parylene-C, "nanocoat" (Chipquick, Niagara Falls, NY, USA), and nitrocellulose-based polymer film (DS—Sally Hansen Diamond Strength Nail Polish). Parylene-C is deposited using a room-temperature low-pressure chemical vapor deposition process using Specialty Coating Systems Parylene-C dimer as a precursor for 0.5 µm and 2.0 µm thicknesses. An adhesion promoter (A-174 Silane from CA Cookson Electronics Co., Altoona, PA, USA) is used as an overnight pretreatment on the sensors receiving a Parylene-C thin film. Liquid nanocoating from Chipquick (nanocoat), model NANOCOAT200UV-2, is coated via the dip-coat method and cured at 60 °C for 10 min. Four dip/cure cycles are applied for a target thickness of ~0.5 µm. The nitrocellulose-based polymer (DS) is applied using a single brush-on application and cured at room temperature for 24 h for a target thickness of 2 µm.

The thickness of the waterproofing materials was measured using a Tencor P-2 stylus profilometer on test structures on cleaned glass slides that had been masked with Kapton$^{TM}$ (3 M) tape.

The sensors are then tested in glycerol/water mixtures of 66%, 77%, 80%, and 92% glycerol-to-water ratios by volume. Tap water with a measured electrical conductivity of 267.5 μs/cm (Orion Star™ A212 Conductivity Benchtop Meter, Thermo Scientific, Waltham, MA, USA) was used to ensure the presence of ions. The electrical conductivity of the 66% glycerol/water mixture was measured to be 5.6 μs/cm. The sensors are actuated with the same conditions as when tested in oil. The sensors are cleaned in DI water and dried between tests.

Long-term reliability tests are performed in 66% glycerol/water mixtures in selected sensors for 1.2 million actuations, which corresponds to ~1.5 days of continuous actuation at a rate of 10 actuations per second. The test conditions are 20 V, 50 μsec pulse at a repetition rate of 10 Hz. Since there were no failures, the actuation energy was increased by 5x to accelerate failures by increasing the heater voltage pulse bias to 45 V. Additional long-term tests were performed with NaCl added to the 66% glycerol/water mixture. The amount of NaCl added was determined to increase the electrical conductivity of the fluid to a measured value of 280 μs/cm, which approximates the initial tap water conductivity

## 3. Results and Discussion

The fabricated thermally actuated viscosity sensors used in this study are based on the work presented in [18]. The working principle of this sensor is based on the thermal impulse actuation of a thin silicon membrane, which vibrates at its natural frequency. When the sensor is immersed in a fluid, the underdamped response of the system can be analyzed to obtain fluid properties such as density and viscosity. The thermal impulse is applied to the membrane via an in situ polysilicon resistor. The movement of the membrane is measured through a Wheatstone bridge of boron-implanted p+ silicon piezoresistors. During normal operating conditions, the center of the membrane has been shown to be displaced by ~120 nm when an instant power of 4 Watts is applied [29,30]. The driving and amplification circuitry are described in detail in [29].

### 3.1. Fabrication and Packaging of Thermally Actuated MEMS Viscosity Sensors

Careful packaging of the fabricated sensors is critical to achieve effective insulation of the metal signal traces from the conductive fluids. Figure 1 shows a cross-section schematic of the sensor material components. As described in detail by [18], a silicon membrane of 15 μm of thickness is fabricated on SOI substrates. Sensing resistors are placed in a Wheatstone configuration by ion-implanting boron ions ($B_{11}^+$) directly into the silicon. These sensors are used to monitor the vibration of the membrane. The membrane actuation is accomplished by pulse-heating the polysilicon heater (0.5 μm thick) in the center of the membrane. Aluminum (0.75 μm thick) is used as the thin film material connecting the actuating and sensing elements. All these layers are insulated from one another using $SiO_2$. In addition, a final 0.5 μm thick $SiO_2$ layer is used as a final passivation layer, burying the interconnecting aluminum. This top $SiO_2$ layer is etched in certain regions to create pad openings to connect the aluminum to the PCB via wire bonds.

Figure 2 shows a top-down picture of a packaged device before testing in air and oil. Text labels are included to indicate the main actuation and sensing components and the exposed metal pads and wire bonds that need to be protected during packaging. Epoxy is applied over the wire bonds and the exposed metal pads for mechanical stability and waterproofing. The center of the sensor is where the actuating/sensing membrane is located and must be kept without thick coatings to ensure its proper actuation and interaction with the fluid under test. The waterproofing material has not been applied at this point.

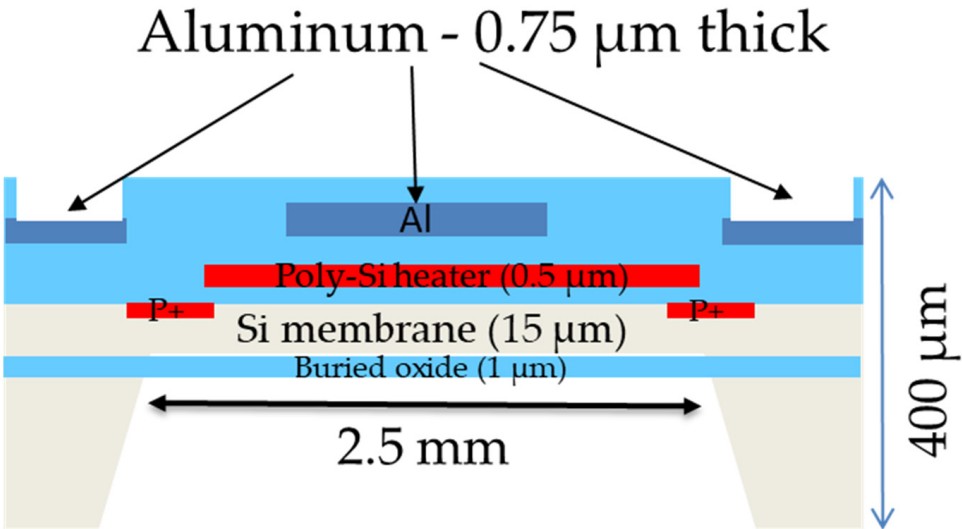

**Figure 1.** Cross-sectional schematic of the basic sensor structure indicating the thickness of the different materials used. Not to scale.

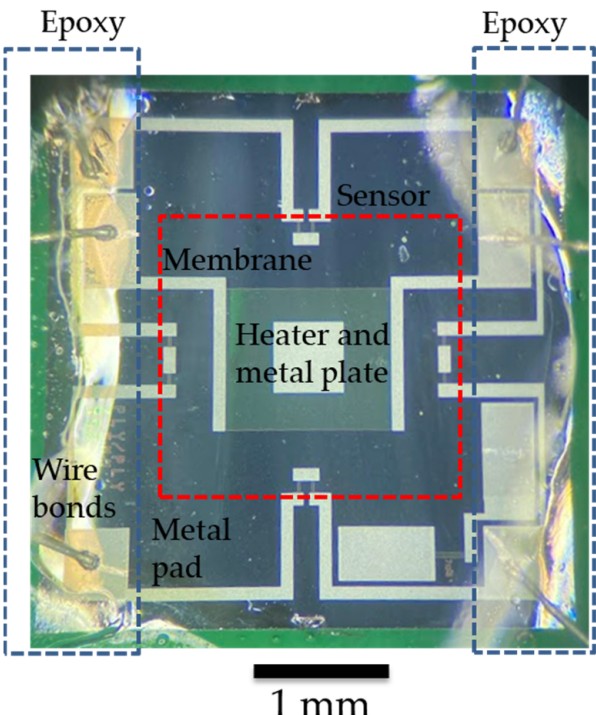

**Figure 2.** Top-down microscope picture of a packaged sensor with labels indicating the main components, areas that need to be protected with epoxy, and the membrane region, which must remain epoxy-free. The shown scale bar is 1 mm.

### 3.2. Test in Oil with Varying Viscosity at Room Temperature

A total of eight sensors were packaged and tested. The initial test measures the natural vibration frequency to verify sensor functionality after packaging. All eight sensors resonated at a frequency between 30 and 32 kHz, indicating full functionality. The sensors were then tested individually in standard oils of different viscosities for 3 min at a measured room temperature of 19 °C, as illustrated in Figure 3. The packaged sensor is completely submerged in a vial containing the fluid being tested. The PCB is connected to the control and amplification electronics with a ribbon cable carrying the actuation and sensing signals as described in [18]. The response of the sensor is monitored with an oscilloscope. The

signal is averaged and analyzed in real time with LabVIEW. The viscosity of the standard oils used were 20.7, 53.5, 86.3, and 330.7 cSt, respectively, for N10, N24, N35, and N100 oils. The sensors were cleaned between tests to avoid cross-contamination of the standard oils. Figure 4 shows a typical output obtained with an oscilloscope while testing a particular sensor in the four different standard oils. As seen in Figure 4, the output response follows the underdamped response of a harmonic oscillator, and measurements can be made on the frequency and decaying exponential from which other characteristic values, such as the damping coefficient or Q factor, can be extracted.

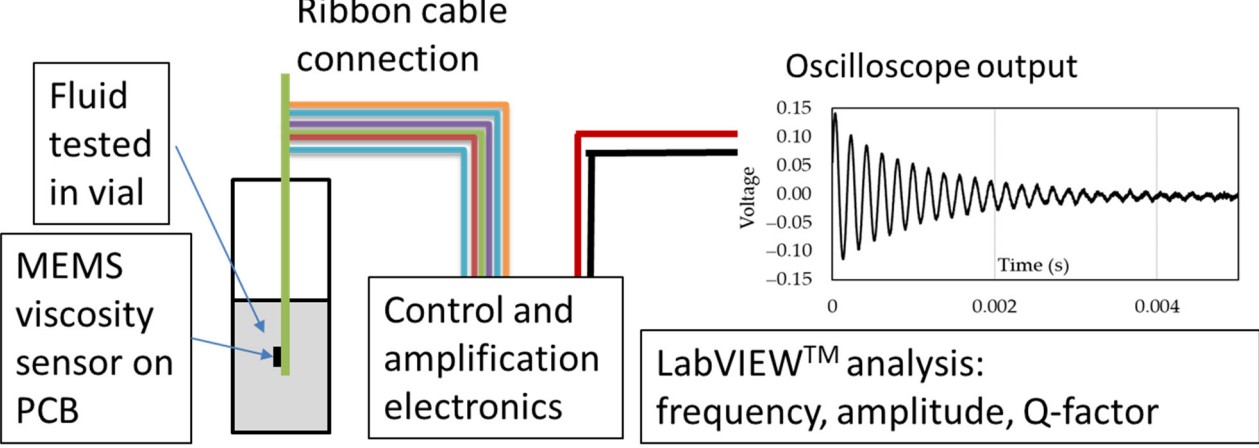

**Figure 3.** Schematic of test setup indicating the placement of the MEMS viscosity sensor on a PCB in a vial with the fluid being tested. Connections are made from the control and amplification electronics to the PCB via a ribbon cable. The output voltage of the sensor is monitored through an oscilloscope, and the signal is analyzed in real time with LabVIEW.

LabVIEW is used to extract the frequency, Q factor, and amplitude of the waveforms for approximately 3 min each at 1 s intervals. Figure 5a shows the typical output collected using LabVIEW for the quality factor of a particular sensor. The Q factor is extracted by finding the decaying exponential [18]. As seen in Figure 5b, the value of Q decreases as the sensor is placed in oils of increasing viscosity. The sensors are carefully placed in the same position during testing. Care must be taken not to introduce bubbles to the back of the diaphragm, as this would result in erroneous measurements and results [31]. It can also be observed from these data that a conditioning time is needed before the measured value reaches a steady-state value.

Average and standard deviations of Q are calculated for each test interval for each particular oil, and the data are plotted against the viscosity to determine the response of each sensor to changes in fluid viscosity. Figure 5b summarizes the data collected for all eight sensors. The average values for Q follow an exponential decay inversely proportional to the kinematic viscosity of the fluid, as predicted in [18] as $Q \approx 1/\xi$ and $\xi = (\upsilon/\omega a^2)^{0.5}$, where $\upsilon$ is the kinematic viscosity of the fluid, $\omega$ the radial frequency, and $a$ is the length of one of the sides of the membrane. As such, $Q \approx \upsilon^{-0.5}$. The variation seen in the behavior from sensor to sensor is due to fabrication and packaging nonidealities of a lab environment. To visually aid in the interpretation of the data and sensor behavior with viscosity, three solid lines are superimposed onto the data of Figure 5b, indicating exponential fits of $\upsilon^{-0.4}$, $\upsilon^{-0.5}$, and $\upsilon^{-0.7}$, respectively.

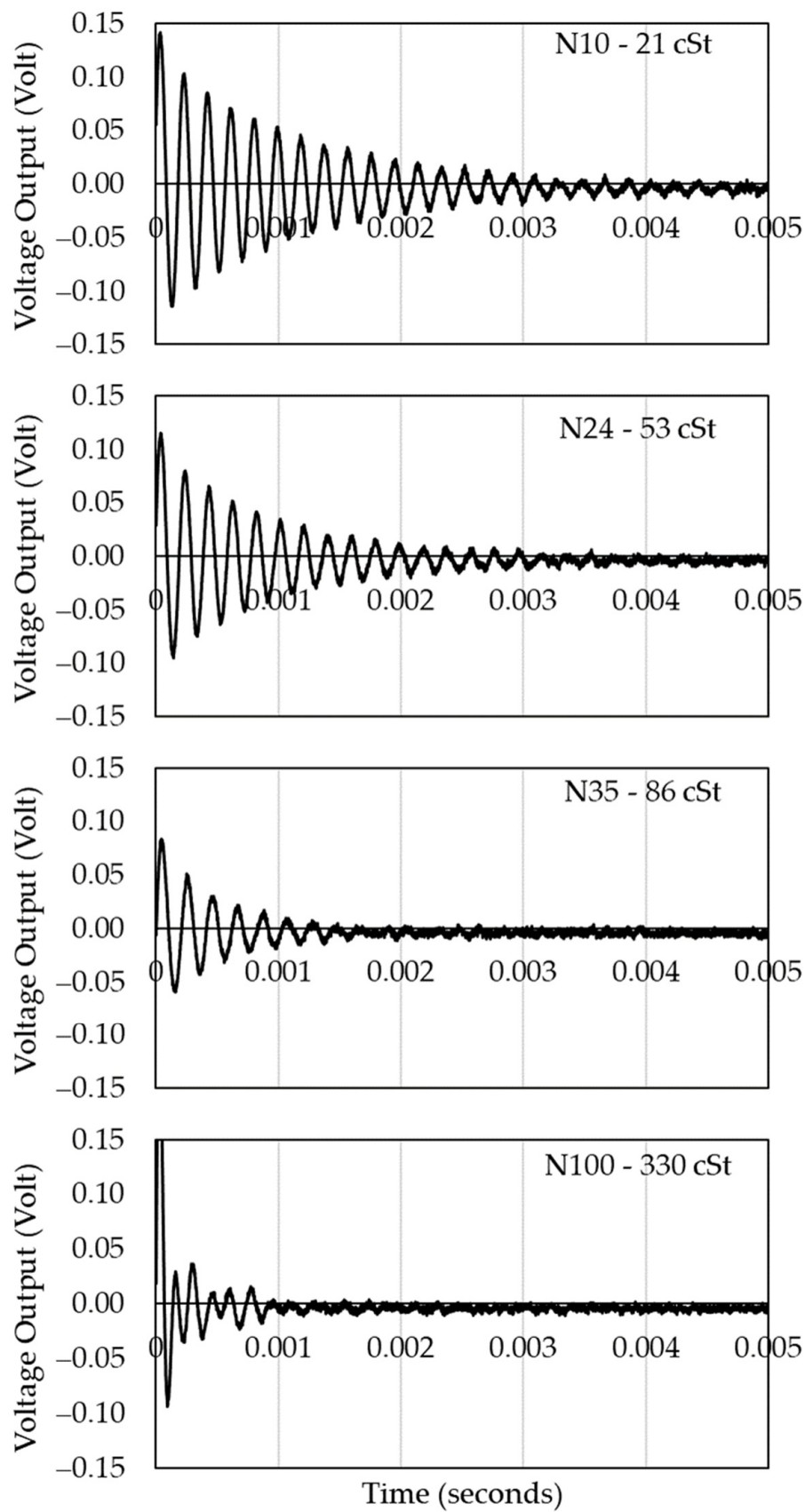

**Figure 4.** Sequence of the sensor's response when interrogating standard oils of different viscosities.

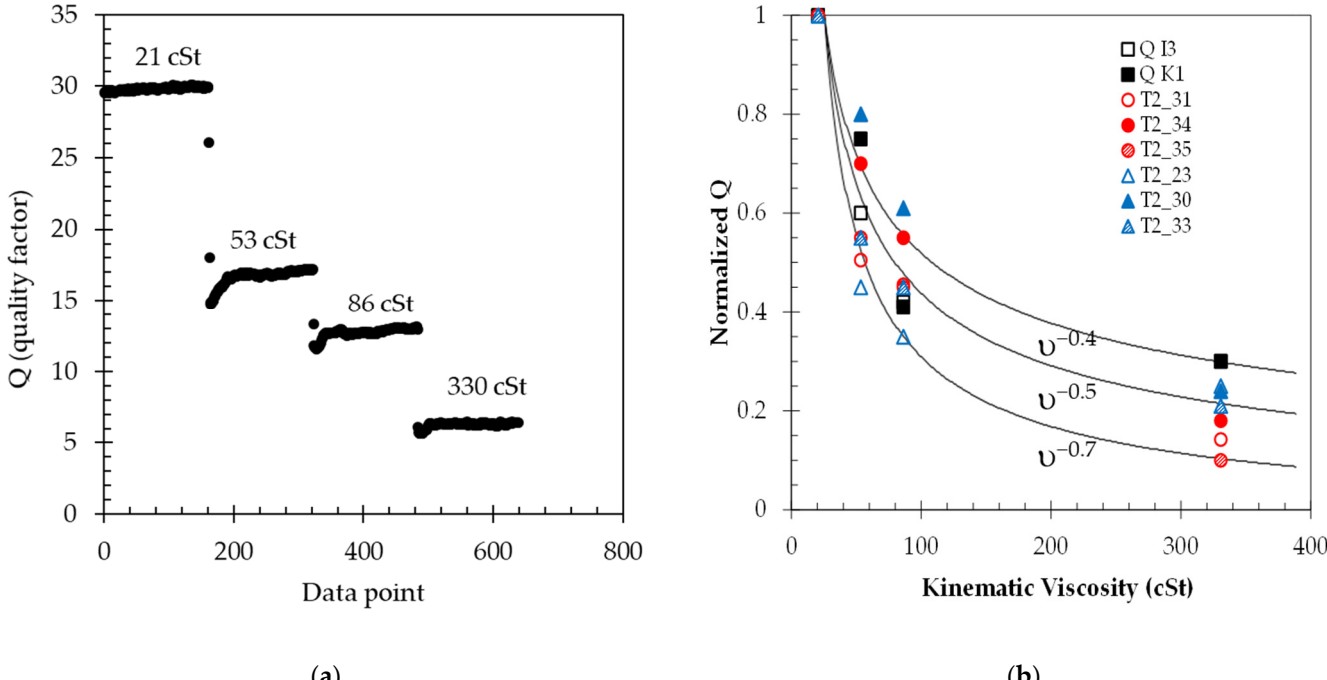

(**a**)                                                                                          (**b**)

**Figure 5.** (**a**) Typical extracted data from the waveform response of a sensor in different oils. Each data point is collected at 1 s intervals; (**b**) Normalized Q-factor value calculated for each sensor in oils of different viscosities. Each data point is an average of about 180 data points. The standard deviation of the measurement is about 1% for the lower viscosities and 3% for the higher viscosities but is not shown for simplicity.

### 3.3. Failure in Water

One sensor was placed in tap water, without a waterproofing coat, and tested with the same test conditions described in the methodology section (20 V, 50 μsec pulse at a repetition rate of 5 Hz). The vibration characteristics were monitored until the device failed. As shown in Figure 6a, the Q of the vibrations was measurable but erratic from the start before failing after ~100,000 actuations (approximately 5 h). Under microscope inspection, it was observed that the metal line that connects to the membrane heater actuator had been corroded, resulting in an open circuit condition. SEM analysis of the failure, as seen in Figure 5b, indicated that a crack or a pinhole in the top $SiO_2$ passivation layer, which is applied during the microfabrication process, led to water reaching the metal line and corroding it through galvanic corrosion [31]. Even though the sensors were fabricated with a top $SiO_2$ passivating layer and functioned in conductive fluids, they provided an unstable output and failed within a few thousand actuations. It is important to keep in mind that this $SiO_2$ passivating layer is not intended to act as a waterproofing material, and failures are expected.

### 3.4. Apply Waterproofing

The sensors were cleaned once more with a degreaser agent, rinsed in DI water and IPA, and dried. The waterproofing material was then applied to the membranes, as described in the methods sections. The objective is to apply a thin layer so that it does not disturb the sensor operation but is effective at protecting the electronic signals from interacting with a conductive fluid. While the CVD deposition of the Parylene-C materials required pretreatment in an adhesion enhancement agent [31], the application of the nanocoat material and nail polish (DS) did not require any additional steps. The resulting thickness of the applied waterproofing coats is shown in Table 1 as the correlation of the sensor number and waterproofing material.

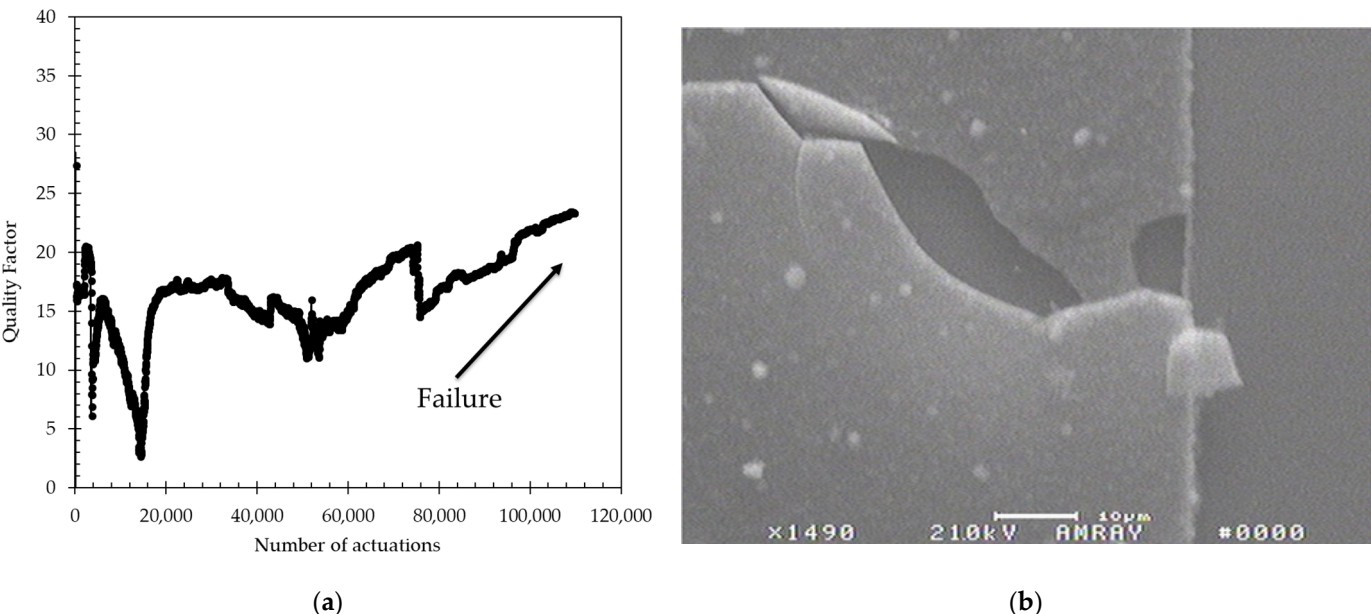

(**a**)                                                (**b**)

**Figure 6.** (**a**) Quality factor as a function of actuation for a sensor tested in DI water without any waterproofing coat. The sensor functions, but it is very variable and fails after approximately 100,000 actuations; (**b**) SEM of location near the failure point where the metal line is discontinuous, resulting in an open circuit.

**Table 1.** Measured thickness of waterproofing materials.

| Sensor | Waterproofing Materials | Measured Thickness |
|---|---|---|
| Q I3 C | Parylene-C (CVD) | 1.93 μm |
| Q K1 C | Parylene-C (CVD) | 0.52 μm |
| T2_31_nano | Nanocoat 4 × (dip coat) | 0.73 μm |
| T2_34_nano | Nanocoat 4 × (dip coat) | 0.91 μm |
| T2_35_nano | Nanocoat 4 × X (dip coat) | 0.86 μm |
| T2_23_DS | DS (brushed on) | 3.2 μm |
| T2_30_DS | DS (brushed on) | 2.7 μm |
| T2_33_DS | DS (brushed on) | 2.6 μm |

The vibration frequency of a simply supported square thin plate can be calculated according to Equation (1):

$$f_{air} = \frac{19.74}{2\pi a^2} \left[ \frac{Eh^3}{12\rho h(1 - n^2)} \right]^{\frac{1}{2}},$$ (1)

where E is the Young's modulus of the material, $a$ is the length of the plate, $h$ is its thickness, $\rho$ is its density, and $n$ is its Poisson's ratio.

As described in [32,33], the effect of viscosity on the free vibration response of thin plates can be regarded as an energy dissipative element that modifies the added virtual mass $\beta$ of a vibrating element in a fluid with density $\rho_{fluid}$ in Equation (1) according to:

$$\omega_{fluid} = \frac{\omega_{air}}{\sqrt{1 + \beta}},$$ (2)

where β is defined as

$$\beta = 0.6538 \frac{\rho_{fluid} a}{\rho_{plate} h} (1 + 1.082\xi),$$ (3)

where the energy dissipation of the system is characterized by $\xi$, which is dependent on the kinematic viscosity $v$, the radial frequency of vibration $\omega$, and the radius $a$ of the membrane

$$\xi = \sqrt{\frac{v}{\omega a^2}}. \tag{4}$$

In addition, the quality factor Q can then be defined as

$$Q = 2\pi \frac{energy_{stored}}{energy_{dissipated_{per_{cycle}}}} \approx \frac{0.95}{\xi} \tag{5}$$

Equation (1) can be expressed as a function proportional to the spring constant $k$ and mass $m$ of the vibrating element:

$$f_{air} \propto \sqrt{\frac{k}{m}} \tag{6}$$

With the addition of another layer, the spring constant $k$ is expected to change by a fraction $x$ related to the spring constant of the material $k_{coat}$, and the mass $m$ increases proportionally to the $m_{coat}$ according to Equation (7).

$$f_{air} \propto \sqrt{\frac{k}{m}} \approx \frac{k_{Si}(1 + x * k_{coat})}{m_{Si} + m_{coat}} \tag{7}$$

Therefore, both the frequency and the Q factor are expected to decrease with the addition of the waterproofing coat as long as the change in mass is more significant than the change in the effective spring constant of the membrane.

The expected changes in frequency Q factor were estimated based on the nominal density and the measured thickness of the coating materials according to Equation (7). To that effect, the nominal densities of 1.3 Kg/m$^3$ for Parylene-C [34], 1.6 Kg/m$^3$ for nanocoat "Chipquick" [35], and 1.7 Kg/m$^3$ for nitrocellulose film [36] were used along the measured thickness shown in Table 1. The mass of the original membrane can be calculated from the bulk density of silicon (2330 kg/m$^3$) and the dimensions (2.5 mm $\times$ 2.5 mm $\times$ 18 µm). As such, and according to Equations (5) and (7), the frequency is expected to decrease between 2% and 13%, whereas the Q factor is expected to decrease between 13% and 36% for the thinnest and thickest films, respectively.

Figure 7 shows a comparison of the before and after frequency and Q values of the sensors that were coated with the different materials when tested in fluids with a kinematic viscosity rating of 20–21 cSt. As seen in Figure 7a, the samples coated with Parylene-C show the least change, with a decrease in resonant frequency between 1% and 4%. The samples coated with the nanocoat show a more significant decrease of 13% to 19%. The samples coated with nail polish (DS) show more inconsistent results, with two samples increasing in frequency by 17% and 25% and one decreasing by 5%. More significant variation is observed in Q, with a reduction of 20% and 50% (Figure 7b). These measured values are in a family with the estimated values based on the added mass to the membrane. The estimated values do not take into account changes to the spring constant of the membrane, as the Young modulus of the tested materials is not readily available.

When the change in frequency and Q are plotted against the measured thickness, as seen in Figure S1, a loose correlation can be observed between the parameters, indicating that the change in frequency and Q is negative for thinner films but turns positive for thicker films. As such, it may be possible to find a thickness for each particular material that could result in no change as the added mass and the effective spring constant are balanced.

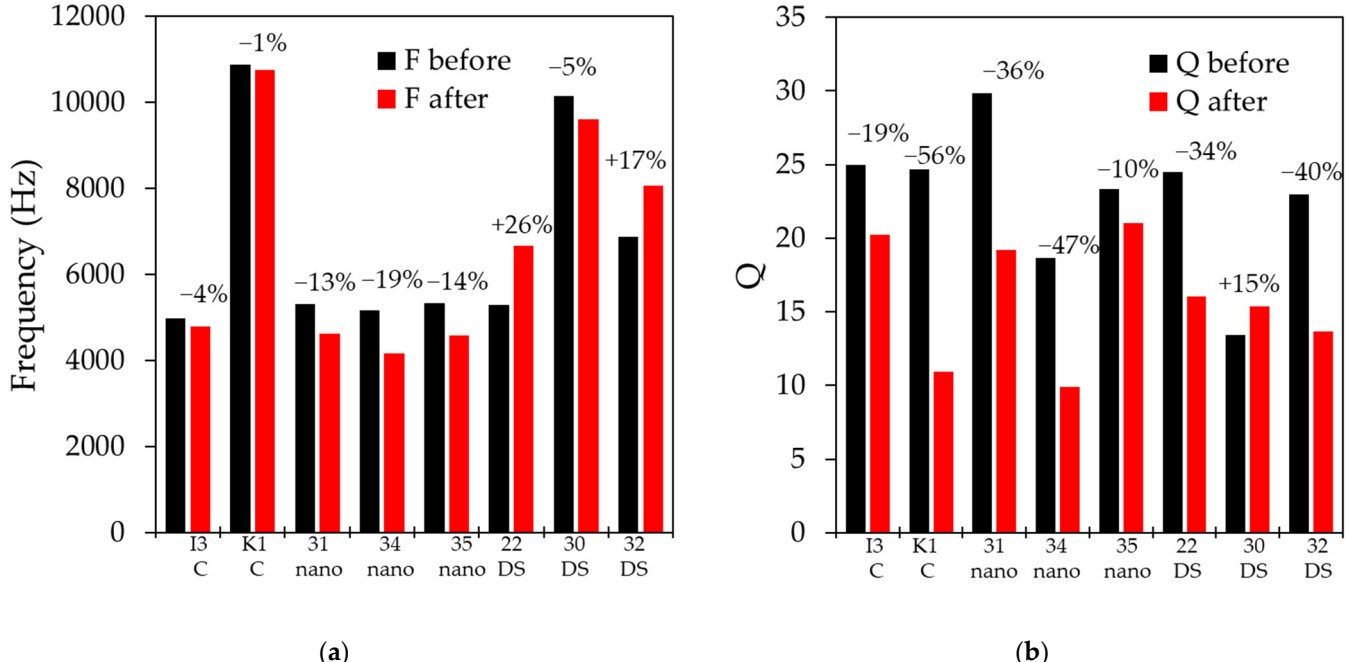

(**a**)                                                                                      (**b**)

**Figure 7.** (**a**) Measured sensor frequency in a viscous fluid (20–21 cSt) before and after the application of the indicated waterproofing material; (**b**) measured sensor Q factor in a viscous fluid (20–21 cSt) before and after the application of the indicated waterproofing material.

### 3.5. Test in Glycerol with Varying Viscosity at Room Temperature

The sensors were then tested in varying solutions of water/glycerol mixtures that were chosen to approximately match the viscosity of standard oils used in the previous tests. The expected viscosities of the 66%, 77%, 80%, and 92%, by volume, of glycerol in water were 21.3, 56.3, 70.4, and 309.0 cSt, respectively. The sensors were actuated with the same test conditions (20 V, 50 μsec pulse at a repetition rate of 5 Hz), and data were collected at 1 s intervals for 3 min.

The normalized averages of the quality factor for each are shown in Figure 8. The value of the quality factor decays in a similar fashion to that of the sensor before the application of the waterproofing material. On the other hand, the measurements taken in the mid-range of viscosities show a larger spread of values when compared to the in-oil measurements due to the additional mass and changes in vibration frequencies, as described in [31]. The standard deviation was also calculated to be around 1% for the lower viscosities and 3% for the higher viscosities. The data of these averages and standard deviations are presented in Tables S1 and S2 in the Supplementary Information for both the tests in oil and glycerol. Solid lines are superimposed onto the data to aid in the visualization of the data, indicating exponential fits of $\upsilon^{-0.4}$, $\upsilon^{-0.5}$, and $\upsilon^{-0.7}$, respectively.

The response of each of the sensors was plotted individually to compare the before and after response of each sensor, which is included in Figures S2 and S3 in the Supplementary Material. Figure S4 shows a log-log version of Figures 5b and 8, where the summary of the fits and their spread can also be visually observed. Table 2 shows the fitting terms, where an increase in the value of the exponential term is seen between each sensor when compared individually before and after the application of the waterproofing material. This variation also correlates with the observation of the data shown in Figures 5b and 8.

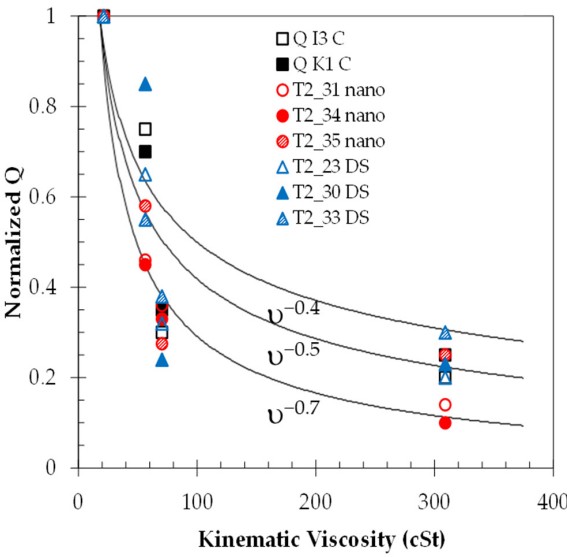

**Figure 8.** Normalized Q-factor value calculated for each sensor in glycerol/water mixtures of different viscosities. Each data point is an average of about 180 data points. The standard deviation of the measurement is about 1% for the lower viscosities and 3% for the higher viscosities but is not shown for simplicity.

**Table 2.** Fitted power model of normalized Q as a function of viscosity.

| Sensor | Before Waterproofing | | | After Waterproofing | | |
|---|---|---|---|---|---|---|
| | Y-Intercept | Exponential Term | Rsq | Y-Intercept | Exponential Term | Rsq |
| Theoretical→ | 4.58 | −0.5 | - | 4.58 | −0.5 | - |
| Q I3 C | 3.49 | −0.44 | 0.91 | 6.47 | −0.62 | 0.817 |
| Q K1 C | 3.90 | −0.46 | 0.85 | 4.71 | −0.53 | 0.84 |
| T2_31 nano | 8.56 | −0.69 | 0.98 | 7.53 | −0.69 | 0.99 |
| T2_34 nano | 10.25 | −0.68 | 0.84 | 13.21 | −0.85 | 0.99 |
| T2_35 nano | 4.00 | −0.44 | 0.96 | 5.14 | −0.58 | 0.82 |
| T2_23 DS | 4.71 | −0.56 | 0.95 | 6.49 | −0.63 | 0.87 |
| T2_30 DS | 5.73 | −0.53 | 0.95 | 5.31 | −0.58 | 0.63 |
| T2_33 DS | 5.34 | −0.56 | 0.99 | 3.35 | −0.45 | 0.85 |

*3.6. Long-Term Testing*

Selected sensors from each of the applied waterproofing coatings were tested for long-term operation in 66% water/glycerol mixtures (electrical conductivity of 5.6 µs/cm). In order to observe the effect of the electrical conductivity of the fluid, one sensor coated with nanocoat was tested in a NaCl + 66% water/glycerol solution (electrical conductivity of 280 µs/cm). Testing under normal operating conditions (20 V, 50 µsec pulse at a repetition rate of 5 Hz) showed that the waterproofed sensors do not fail as the membrane movement is restricted to the elastic regime. Figure 9 shows the value of the Q value for the three different waterproofing coats in the tested solutions. No failures are seen over 1.2 million actuations, which would correspond to 2.7 days of continuous operation at 5 actuations per second. Other possible operating conditions that are not continuous could require, for example, daily measurements of 5 min, extending the tested 1.2 million actuations to >2 years.

In order to accelerate the failures, the energy delivered through the heater actuator was increased by ~5x over the normal operation conditions (from 20 V to 45 V) in 66% water/glycerol mixtures. This ~5x increase in power and energy results in an expected 5x increase in the displacement at the center of the membrane from 120 nm to 600 nm [36]. With these accelerated conditions, it is seen in Figure 10 that the Parylene-C failed after approximately 100 min, the nanocoat samples failed after 3 h, and the nail polish (DS) sample failed after 58 min. Microscope inspections indicate that galvanic corrosion has

taken place in the heating element metal trace for all samples, starting at a location near the edge of the membrane, as shown in Figure 11a,b.

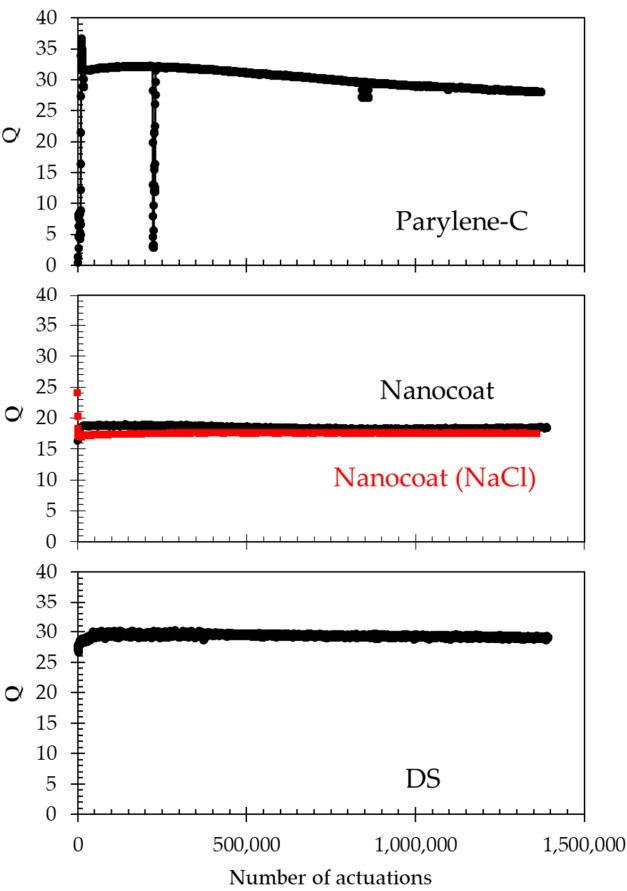

**Figure 9.** Q factor measured for selected sensors with the indicated waterproofing film. Black data were obtained in 66% water/glycerol mixtures (electrical conductivity of 5.6 µs/cm). The sample label Nanocoat (NaCl) was tested in a NaCl + 66% water/glycerol solution (electrical conductivity of 280 µs/cm). These sensors did not fail after more than 1.2 million actuations.

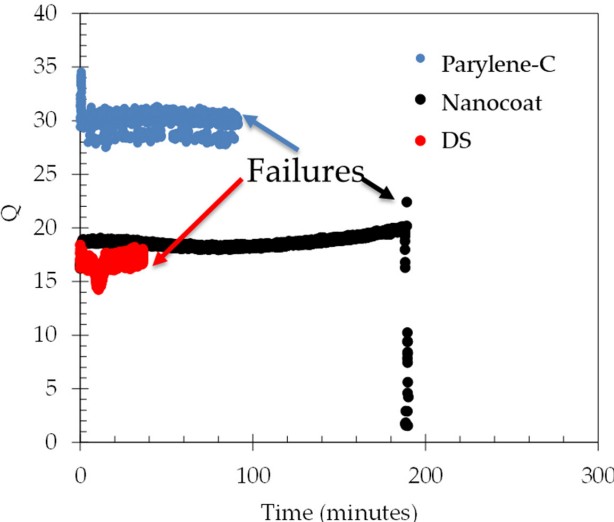

**Figure 10.** Q factor measured for selected sensors with the indicated waterproofing film. These three sensors were tested under high bias (5x normal bias) to accelerate the failures. The nanocoat film lasted the longest, approximately 3 h.

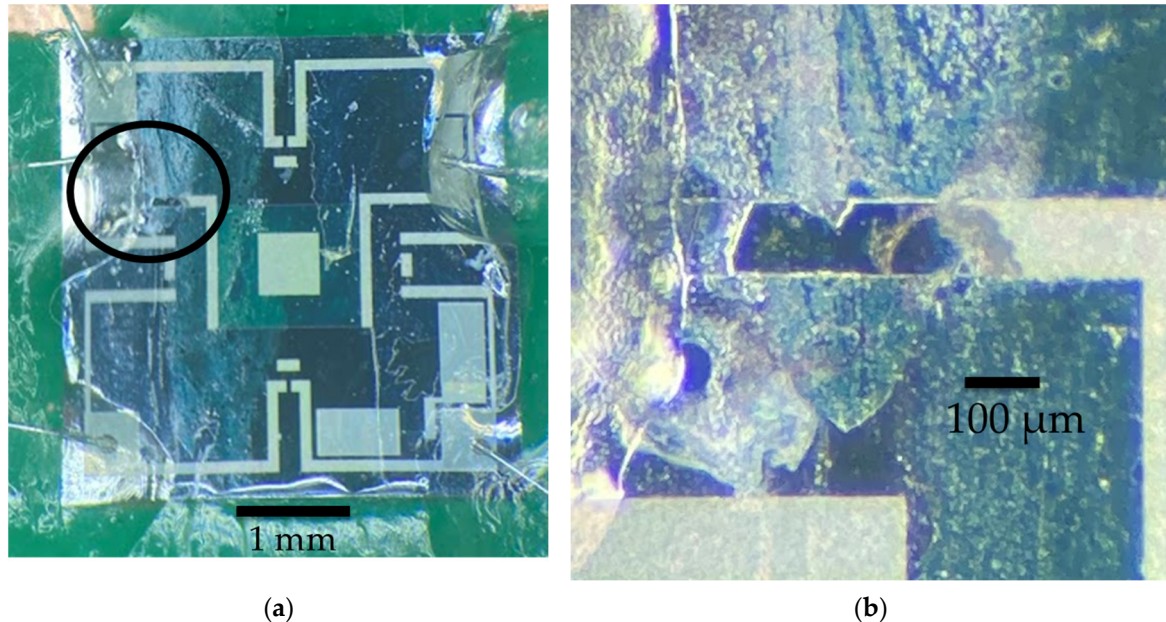

**Figure 11.** (**a**) Microscope picture of a device coated with nanocoat, which failed after nearly three hours of actuation under accelerated bias conditions that were 5x higher than normal operation conditions. The black circle highlights the location of the observed failure; (**b**) detail of the failure location on the metal line that connects the heater in the center of the membrane. Galvanic corrosion is observed.

The nanocoat material seems to provide longer-lasting protection than the other two materials when tested under harsh test conditions. The chemical composition of the nanocoat material is proprietary as it is a commercial product used to protect PCB components with a maximum operating temperature of up to 175 °C, but according to its datasheet, it contains fluorocarbons and fluoroacrylates and no silicone additives [37]. Fluorocarbons have been shown to form a hydrophobic polymer [38]. As such, the application of the nanocoat material to glass has shown a water contact angle of 113.3° and an oil contact angle of 82.0° [37]. Parylene-C provided adequate protection during normal operating conditions but seemed to mechanically fail earlier than the nanocoat material when the actuation magnitude was increased. Finally, the nail polish material (DS) showed adequate protection during normal operating conditions but failed the earliest when the conditions were increased. Overall, the results presented here indicate that adding a protective coating to a micro-actuator will significantly improve its reliability in electrically conductive fluids.

## 4. Conclusions

Parylene-C, fluoroacrylate-based "nanocoat" (Chipquick) polymer, and nitrocellulose-based polymer film (DS—Sally Hansen Diamond Strength Nail Polish) were coated as thin layers of waterproofing materials on different thermally actuated viscosity sensors. All three coating materials provided adequate protection when they were used to waterproof the sensors under normal operating conditions. Although the vibration response of the sensors was modified, it did not affect their functionality in a significant way when measuring conductive fluids based on glycerol/water mixtures. All the sensors that were treated lasted over 1.2 million actuations without any decay in performance or failures. When the test conditions were increased to accelerate failures, the "nanocoat" samples lasted 2x longer than the other two due to their more stable chemical composition and lower stiffness. Visual analysis of the failures indicates that the edge of the diaphragm, which undergoes the largest stress and strain values during actuation, was the main location of the mechanical failure. This work guides coating solutions for microscale actuators operating in harsh and damaging environments.

**Supplementary Materials:** The following supporting information can be downloaded at: https://www.mdpi.com/article/10.3390/act13020057/s1, Figure S1: Change in frequency and Q vs. added film thickness for the different coat materials. Figure S2: Fitted exponential to the normalized Q factor versus kinematic viscosity response, which was measured on sensors before and after applying a Parylene-C coating. Figure S3: Fitted exponential to the normalized Q factor versus kinematic viscosity response, which was measured on sensors before and after applying (a) nano coat polymer (b) nail polish DS coat. Figure S4: Log-log plots of the normalized q-factor value calculated for each sensor in (a) standard oils and (b) glycerol/water mixtures of different viscosities. Each data point is an average of about 180 data points. The standard deviation of the measurement is about 1% for the lower viscosities and 3% for the higher viscosities, but not shown for simplicity. Table S1: Average and standard deviation of the characteristic vibration properties of the tested viscosity sensors in different oils.

**Author Contributions:** Conceptualization, L.G., S.C., K.R. and I.P.; methodology, C.L., L.J. and A.U.; software, S.C., C.L., L.J., A.U. and I.P.; validation, L.G., S.C., K.R. and I.P.; formal analysis, L.G., K.R. and I.P.; investigation, L.G., S.C., K.R., C.L., L.J., A.U. and I.P.; resources, I.P.; data curation, L.G., S.C., K.R., C.L., L.J., A.U. and I.P.; writing—original draft preparation, L.G. and I.P.; writing—review and editing, A.U., L.G. and I.P.; visualization, L.G., S.C., K.R., C.L., L.J., A.U. and I.P.; supervision, I.P.; project administration, I.P.; funding acquisition, I.P. All authors have read and agreed to the published version of the manuscript.

**Funding:** This research was partially funded by Poseidon Systems LLC, 830 Canning Pkwy, Victor, NY 14564, USA.

**Data Availability Statement:** Data are available upon request to ixpeme@rit.edu.

**Acknowledgments:** The authors acknowledge support from the Kate Gleason College of Engineering at Rochester Institute of Technology.

**Conflicts of Interest:** The authors declare no conflicts of interest.

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
