# Peer review of "Waterproofing a Thermally Actuated Vibrational MEMS Viscosity Sensor"

_actuators, doi:10.3390/act13020057_

Round 1

Reviewer 1 Report

Comments and Suggestions for Authors

This manuscript entitled "Waterproofing a thermally-actuated vibrational MEMS viscosity sensor" investigates the effect of waterproof coating on the measurement performance of vibrational viscosity sensor. Theoretical background and experimental investigation are well written in the manuscript. The authors tested several coating materials with different thicknesses, and concluded that fluoroacrylate-based “nano coat” (Chipquick) polymer showed the best tolerance and performance for the viscosity measurement in a wide range of viscosity. Although there is a significant unclear mechanism of the shift in Q-factor after coating, the trend in the most case seems to be reasonable. However, an improvement of the sensor for conductive solution needs the investigation of various conductivity. This manuscript did not mention the conductivity of the solution. Consequently, I would like to accept this manuscript in Actuators after addressing the following items.

[Major]

- The authors can estimate the shift of Q-factor by coating using nominal information of density of the coating materials and measurement thickness. Please mention this fact and comparison, and describe why it is difficult to estimate the shift.

- I would like to recommend the authors to conduct investigation with different conductivities with reliable measurement information. Can you perform the similar test with similar viscosity but with different conductivity? For example, the use of NaCl in the solution may easily alter only the conductivity. Then the ability of the electrical isolation in the sensor can be evaluated.

[Minor]

- L65 [Puchades] seems to be one of the authors' previous reference. Please clarify this.

- In the manuscript, please unify the expression of SiO2. The manuscript  has the expression of SiO2 in several parts.

- L185-186 Do you distinguish "kinetic viscosity" and "kinematic viscosity"? And why do the authors use "x" for kinematic viscosity although they use "v" in Equation. 

Reviewer 2 Report

Comments and Suggestions for Authors

Report on paper "Waterproofing a thermally-actuated vibrational MEMS viscosity sensor" submitted by Gan et al., for publication in Actuators (actuators-2790209).

The authors investigated experimentally waterproofing a vibrational MEMS viscosity sensor. Although the topic of the paper is interesting, its originality and goal are not well highlighted. It cannot be accepted in its present form and the authors must perform several modifications by addressing the following comments:

1.    The abstract should be rewritten in a more compact and precise way while highlighting the research originality and goal.

2.    The introduction should be extended while enriching the state of the art related to MEMS viscosity sensors.

3.    Can the authors provide a figure illustrating the experimental set-up?

4.    In section 3, the limitations of the obtained results should be discussed from a critical point of view.

5.    Please check figure numbering (there are two figures 7 and two figures 8) and make sure all figures are cited in the paper.

6.    The quality of several figures should be enhanced.

7.    Please remake Table 2 in a more professional way.

Comments on the Quality of English Language

The authors should carefully check the English writing as well as the syntax errors in the whole paper.

Round 2

Reviewer 1 Report

Comments and Suggestions for Authors

The revised manuscript have addressed all the comments well, so I would like to recommend this manuscript to be published in Actuators.

Author Response

Thank you for your review!

Reviewer 2 Report

Comments and Suggestions for Authors

The authors could address the following minor points:

1- In line 134, Figure 1 instead of Figure 1a.

2- The quality of figures 5 and 8 should be enhanced.

Comments on the Quality of English Language

The authors should carefully check the English writing as well as the syntax errors in the whole paper.
